# The Association between IL-1β and IL-18 Levels, Gut Barrier Disruption, and Monocyte Activation during Chronic Simian Immunodeficiency Virus Infection and Long-Term Suppressive Antiretroviral Therapy

**DOI:** 10.3390/ijms25168702

**Published:** 2024-08-09

**Authors:** Siva Thirugnanam, Chenxiao Wang, Chen Zheng, Brooke F. Grasperge, Prasun K. Datta, Jay Rappaport, Xuebin Qin, Namita Rout

**Affiliations:** 1Division of Microbiology, Tulane National Primate Research Center, Covington, LA 70433, USA; thiru@tulane.edu; 2Department of Microbiology and Immunology, Tulane University School of Medicine, New Orleans, LA 70112, USA; cwang27@tulane.edu (C.W.); chenzhengsx@gmail.com (C.Z.); pdatta@tulane.edu (P.K.D.); jrappaport@tulane.edu (J.R.); xqin2@tulane.edu (X.Q.); 3Division of Pathology, Tulane National Primate Research Center, Covington, LA 70433, USA; 4Division of Veterinary Medicine, Tulane National Primate Research Center, Covington, LA 70433, USA; bgrasper@tulane.edu; 5Tulane Center for Aging, Tulane University School of Medicine, New Orleans, LA 70112, USA

**Keywords:** IL-18, IL-1β, IFABP, SIV, immune activation, monocyte

## Abstract

HIV-induced persistent immune activation is a key mediator of inflammatory comorbidities such as cardiovascular disease (CVD) and neurocognitive disorders. While a preponderance of data indicate that gut barrier disruption and microbial translocation are drivers of chronic immune activation, the molecular mechanisms of this persistent inflammatory state remain poorly understood. Here, utilizing the nonhuman primate model of Human Immunodeficiency Virus (HIV) infection with suppressive antiretroviral therapy (ART), we investigated activation of inflammasome pathways and their association with intestinal epithelial barrier disruption (IEBD). Longitudinal blood samples obtained from rhesus macaques with chronic SIV infection and long-term suppressive ART were evaluated for IEBD biomarkers, inflammasome activation (IL-1β and IL-18), inflammatory cytokines, and triglyceride (TG) levels. Activated monocyte subpopulations and glycolytic potential were investigated in peripheral blood mononuclear cells (PBMCs). During the chronic phase of treated SIV infection, elevated levels of plasma IL-1β and IL-18 were observed following the hallmark increase in IEBD biomarkers, intestinal fatty acid-binding protein (IFABP) and LPS-binding protein (LBP). Further, significant correlations of plasma IFABP levels with IL-1β and IL-18 were observed between 10 and 12 months of ART. Higher levels of sCD14, IL-6, and GM-CSF, among other inflammatory mediators, were also observed only during the long-term SIV + ART phase along with a trend of increase in the frequencies of activated CD14^+^CD16^+^ intermediate monocyte subpopulations. Lastly, we found elevated levels of blood TG and higher glycolytic capacity in PBMCs of chronic SIV-infected macaques with long-term ART. The increase in circulating IL-18 and IL-1β following IEBD and their significant positive correlation with IFABP suggest a connection between gut barrier disruption and inflammasome activation during chronic SIV infection, despite viral suppression with ART. Additionally, the increase in markers of monocyte activation, along with elevated TG and enhanced glycolytic pathway activity, indicates metabolic remodeling that could fuel metabolic syndrome. Further research is needed to understand the mechanisms by which gut dysfunction and inflammasome activation contribute to HIV-associated metabolic complications, enabling targeted interventions in people with HIV.

## 1. Introduction

Antiretroviral therapy (ART) with efficient, well-tolerated combinational antiretroviral drug regimens has remarkably increased the life span of human immunodeficiency virus (HIV)-infected individuals worldwide [1]. However, people with HIV (PWH) remain at a high risk of non-AIDS-defining comorbidities, including neurocognitive, metabolic, and cardiovascular diseases (CVDs) [2,3,4], potentially due to persistent immune activation and chronic inflammation. CVD is a major cause of morbidity and mortality in PWH in the ART era [3,4]. The association between immune activation markers, such as serum IL-6 and IL-1β, and indicators of cardiovascular events in both the general population and HIV-1 infected individuals suggests a connection between immune activation and CVDs [5]. Chronic inflammatory processes involving both innate and adaptive immunity are key drivers of CVD pathogenesis, as they alter vascular biology and promote accelerated atherogenesis, plaque formation, and plaque instability [6]. Pro-inflammatory markers, IL-6, and D-dimer are associated with increased mortality in PWH on ART, suggesting that interrupting therapy might further exacerbate this risk [7]. Elevated IL-6 levels correlate with increased tight junction (TJ) permeability in intestinal epithelia (leaky gut), potentially linking it to adverse outcomes like cardiovascular events [8]. These findings align with the emerging evidence suggesting a connection between gut microbial translocation and increased cardiovascular risk in PWH [9].

We have previously demonstrated that epithelial barrier disruption precedes the development of a distinct plasma inflammatory signature during chronic SIV infection with long-term suppressive ART in rhesus macaques [10]. Pro-inflammatory cytokines, such as TNF-α, IFN-γ, IL-18, and IL-1β, can trigger cell-death pathways like apoptosis in macrophages and foam cells, leading to the release of their cholesterol-rich contents into the arterial intima [11]. Furthermore, productive and latent HIV infection activates monocytes/macrophages and mediates an array of molecular signaling pathways with established pathogenic roles in traditional atherosclerosis [3]. One pathway of interest is inflammasome formation and caspase-1 activation [3,12]. Pathogens, stress, and endogenous danger molecules can activate caspase-1 through cleavage or activation by inflammasome formation. Active caspase-1 then cleaves the pro-inflammatory cytokines IL-1β and IL-18 into their active forms, initiating an immune response, pathogen clearance, and systemic tissue inflammation [12,13]. HIV infection stimulates the formation of inflammasome in human monocytes/macrophages and microglia, leading to caspase-1 activation and the release of active IL-1β and IL-18 [14,15,16]. Additionally, activated levels of caspase-1 from inflammasome cleavage increase rapidly during early HIV infection [17]. Caspase-1 activation in HIV and SIV infections triggers pyroptosis, an inflammatory cell-death process that depletes CD4+ T cells and fuels chronic inflammation [12,18,19,20,21]. We have previously reported the presence of caspase-1^+^ macrophages in plaques of PWH [22] and demonstrated elevated serum IL-18 in PWH that correlated with total segments and the number of non-calcified inflammatory plaques [22]. Further, SIV infection in rhesus macaques without ART was shown to increase caspase-1 activation and secretion of IL-18 [23]. However, the exact role of inflammasome activation in monocytes and macrophages during HIV atherogenesis has not been rigorously explored in a clinically relevant model of chronic HIV infection with ART.

Using the rhesus macaque model of chronic HIV infection with suppressive ART, we have previously shown that the emergence of systemic inflammation following short-term recovery with ART is linked to IEBD and loss of gut mucosal immune functions [10]. In this report, we utilized plasma and PBMC samples from SIV-infected macaques under long-term ART for up to 12 months to further explore the association between gut barrier disruption and inflammasome activation and their potential relationship with monocyte activation. Our findings suggest that gut barrier disruption correlates with inflammasome activation during chronic SIV infection despite long-term virus-suppressive ART. This is associated with changes in intermediate monocyte subpopulations and increased proinflammatory cytokines as well as elevated triglycerides in the blood.

## 2. Results

### 2.1. Increased Gut Permeability during Chronic SIV with ART Aligns with Elevated Levels of IL-1β and IL-18

Six adult rhesus macaques were infected via the intra-rectal route with SIV_mac251_ to represent mucosal HIV infection. Following the establishment of set-point viremia, all animals were treated daily with a three-drug combination antiretroviral regimen consisting of Tenofovir Disoproxil Fumarate (TDF), Emtricitabine (FTC), and Dolutegravir (DTG) (Figure 1A). Peak plasma viremia was observed at 2 weeks post-SIV infection at an average of 7.2 log10 copies/mL and reduced to set-point viremia at ~5.8 log10 copies/mL by 4 weeks (Figure 1B). Stable suppression of viremia below the limit of detection (LOD) of the assay (83 viral copy Eq/mL) was achieved in all six animals by 2 months post-ART initiation (Figure 1B).

The plasma levels of inflammasome activation cytokines, IL-1β and IL-18, were found to be significantly increased during acute SIV infection prior to ART (mean values of 9.1 pg/mL and 64.3 pg/mL, respectively; Figure 2A,B). Following ART initiation, IL-1β levels returned to baseline and were maintained at lower levels until at least 5 months after the beginning of the ART (Figure 2A).

ART suppression of viremia, however, failed to reduce IL-18 levels below those observed during the acute phase of infection beyond the initial control at 3 months of ART (Figure 2B). Notably, much higher increases in IL-1β and IL-18 levels were observed in the SIV-infected macaques at 10–12 months of ART, as evidenced by 6-fold higher IL-1β levels and 3.6-fold higher IL-18 levels than the levels observed during acute SIV infection, which were indicative of more robust and persistent inflammasome activation during long-term SIV + ART (Figure 2A,B). As previously reported [10], plasma biomarkers of IEBD, including IFABP and LBP, transiently increased during acute SIV infection and resolved during short-term ART, only to return to significantly higher levels during chronic SIV infection around 8 months of viral suppression with continuous ART (Figure 2C,D). Interestingly, the increase in plasma IFABP and LBP levels preceded the emergence of IL-18 and IL-1β that was observed later at 10 months of ART. Moreover, a correlation analysis of plasma IFABP and LBP with the two inflammasome-related cytokines revealed a significant correlation of both IL-1β and IL-18 with the leaky gut biomarker IFABP between 10 and 12 months of ART (Figure 2E–H), suggesting a link between IEBD and inflammasome activation during chronic SIV + ART.

We next assessed circulating caspase-1 levels as upstream mediators of IL-18 and IL-1β secretion. Although there was an increase in serum caspase-1 at 1 month post-SIV infection (Figure 3A), it did not reach significantly higher levels, unlike our earlier observation in the CD8-depletion model of accelerated SIV pathogenesis with ART [23]. Suppression of viremia during short-term ART in our study lowered serum caspase-1 levels in 5/6 macaques, as evidenced at the 5-month post-ART time-point (Figure 3A) suggesting that early caspase-1 activity at a systemic level was mainly driven by uncontrolled viral replication during acute SIV infection.

In contrast to the highly significant increases in IL-1β and IL-18 levels at the 10-month ART time-point, serum caspase-1 levels this time-point did not increase significantly from the short-term ART time-point at 5 months (Figure 3A). Nonetheless, caspase-1 levels displayed a significant positive correlation with IFABP between 5 and 12 months of chronic SIV infection with ART (Figure 3B), further supporting the role of IEBD in driving inflammasome activation during the late chronic phase of SIV + ART. Since IEBD and microbial translocation are established contributors to systemic immune activation of HIV/SIV infection, we further examined the relationship between sCD14 levels and plasma caspase-1 in our cohort. Unlike IFABP, sCD14 levels were not significantly correlated with caspase-1 (Figure 3C), suggesting that serum caspase-1 levels are more strongly associated with intestinal permeability caused by enterocyte damage than translocation of microbial products. Overall, our data suggest that although ART attenuates circulating caspase-1 levels, IEBD continues to activate the inflammasome pathway during chronic SIV infection with long-term ART.

### 2.2. Elevated Markers of Monocyte Activation and Correlation of Intermediate Monocytes with IEBD during Chronic SIV Infection with Long-Term ART

We have previously reported that the signature of proinflammatory cytokines during chronic SIV infection with ART is distinct from acute phase cytokines and is independent of viral suppression [10]. Among the pro-inflammatory cytokines, we observed significantly higher levels of multiple factors associated with monocyte activation. Furthermore, sCD14, a biomarker of microbial translocation produced by activated monocytes upon LPS stimulation, was found to be significantly higher in plasma samples at the 12-month post-SIV with 8-month ART time-point than the levels at pre-SIV baseline or 1-month post-SIV infection (Figure 4A). Additionally, we observed elevated levels of plasma IL-6, TNF-*α*, and MCP-1 at 12 months of SIV + ART in comparison to baseline levels (Figure 4B–D). Notably, the increase in IL-6 and TNF-*α* was significant in comparison to both baseline and acute SIV infection levels (*p ≤* 0.0007 and *p ≤* 0.002, respectively). Plasma MCP-1 levels showed inconsistent increases during acute SIV infection in two out of six macaques (Figure 4D). The increase in MCP-1 was, however, significant during the chronic phase after 12 months of SIV + ART, with higher-than-baseline levels observed in all individuals within the group (Figure 4D).

The elevated levels of plasma IL-6 and sCD14 suggest that monocyte activation may contribute to the increase in systemic inflammation during long-term SIV + ART despite effective suppression of viremia. We compared the changes in the frequency of monocyte subpopulations in PBMC from baseline to acute SIV infection before ART, and at 12 months post-SIV with ART. A significant increase in the circulating frequencies of activated, proinflammatory CD14^+^CD16^+^ monocyte subsets (intermediate monocytes) was observed during acute SIV infection (Figure 4E). Although long-term ART lowered the frequencies of intermediate monocytes in blood, they remained slightly higher than at baseline. The frequencies of classical monocytes (CD14^+^CD16^−^) and nonclassical monocytes (CD14^lo/neg^CD16^++^) did not differ significantly between pre-SIV and post-SIV timepoints (Figure 4F,G). Notably, the frequencies of intermediate monocytes at 12 months post-SIV displayed a near-significant correlation with plasma IFABP levels (Figure 4H), suggesting an association between IEBD and the activation of intermediate monocytes during chronic SIV infection with long-term ART.

### 2.3. Blood Triglycerides and Glycolytic Capacity in PBMCs during Chronic SIV Infection with Long-Term ART

Studies have shown a strong association between high triglyceride (TG) levels and increased risk of CVD events like heart attack and stroke in HIV-infected individuals [24]. This is likely driven by chronic HIV-associated inflammation and the metabolic effects of certain antiretroviral drugs [25]. Since markers of microbial translocation and TG are positively correlated in HIV-infected patients with suppressed viral replication [26,27], we assessed the longitudinal changes in blood TG levels in our cohort from pre-SIV infection baseline, through acute infection prior to ART, and during long-term suppressive ART. Although blood TG levels were not significantly altered with acute SIV infection, the levels were found to gradually increase from the 5-month ART up to at least the 12-month ART time-point (Figure 5A). Together with increases in IEBD and microbial translocation biomarkers, this increase in TG suggests that SIV-infected macaques under long-term ART may display similar mechanisms of dyslipidemia to those observed in HIV-infected patients undergoing ART.

As HIV-infected individuals display dysfunctional glucose metabolism in monocytes and T cells, which is linked to immune activation and inflammatory disease progression [28], we next assessed the effect of oxidized low-density lipoprotein (Ox-LDL) on aerobic glycolysis in PBMCs. PBMCs derived from control SIV-naïve macaques and our cohort animals after 12 months of SIV + ART were examined for glycolytic capacity (a measure of the maximum rate of conversion of glucose to pyruvate or lactate in a cell) using SeaHorse-based extracellular flux measurements. Our results demonstrated that Ox-LDL enhanced aerobic glycolysis, specifically the glycolytic capacity in PBMCs in both SIV-naïve and SIV + ART groups, by showing an increase in the extracellular acidification rate (ECAR), indicating an increased rate of glycolysis. The glycolytic capacity was enhanced ~3-fold in PBMCs from SIV-infected ART-treated RMs compared to PBMCs from SIV-naïve RMs (Figure 5B). Treatment with Ox-LDL further enhanced the ECAR by ~3.7-fold in PBMCs from SIV-infected ART-treated macaques compared to PBMCs from SIV-naïve macaques (Figure 5B), indicating increased glycolysis in PBMCs during chronic SIV infection with long-term ART.

## 3. Discussion

Gut barrier breakdown and persistent inflammation are hallmarks of chronic HIV infection despite long-term treatment with antiretroviral drugs. In the present study, we explored the potential role of chronic SIV-associated gut barrier disruption and inflammasome activation as drivers of increased monocyte activation and cardiometabolic risk. We provide evidence of a significant correlation between the increased IEBD biomarker IFABP and circulating inflammasome byproducts, IL-1β and IL-18, during chronic SIV infection with long-term suppressive ART. Furthermore, we report increased plasma markers of monocyte activation, along with elevated blood triglyceride levels and enhanced glycolytic activity in PBMCs of macaques with long-term SIV + ART.

IL-1β and IL-18, which are converted into their mature forms by the intracellular cysteine protease caspase-1, are two related cytokines with critical roles in inflammation via the inflammasome pathway. Previous studies have shown that HIV can trigger pattern recognition receptors (PRRs) such as Toll-like receptors (TLRs) [29,30] as well as the inflammasomes [15], leading to secretion of IL-1β and IL-18. We observed that SIV infection similarly triggers increased secretion of IL-1β and IL-18 cytokines into blood and that this activation was even higher during chronic SIV infection despite long-term ART suppression of viremia. This agrees with clinical studies showing that elevation in circulating levels of IL-18 and IL-1β is particularly notable in later stages of HIV infection [31,32]. Our findings of a strong positive correlation between IFABP and both IL-1β and IL-18 levels, as well as caspase-1 during chronic SIV infection with long-term ART, further highlight the link between leaky gut and inflammasome activation that are potential drivers of chronic inflammation. With regard to our findings, Feria et al. have shown that HIV-progressors exhibit higher expression of the inflammasome genes IL-1β, IL-18 and caspase-1 in gut-associated lymphoid tissue (GALT) and PBMCs [33]. However, despite higher plasma IL-1β and IL-18 levels in our cohort at 1-year post-SIV, and in contrast with a previous report showing increased plasma caspase-1 levels after 1 year of untreated HIV-1 infection [17], we did not find significant increase in caspase-1 levels in our cohort. This could be due to a combination of factors, including effective viral suppression with ART and the limitation of our study in measuring plasma caspase-1 instead of direct expression levels in T cells and myeloid cells (monocytes/macrophages) in circulation and tissues. Supporting this, our earlier studies have shown that ART partially attenuates an increase in caspase-1 in circulation but not in lymphoid tissues [23], suggesting that tissue immune activation persists despite adequate suppression of viremia.

High levels of IL-18 and IL-1β in HIV-infected individuals are strongly associated with monocyte activation, contributing to chronic immune activation and inflammation. Our results showing elevated plasma levels of sCD14, IL-6, and TNF-α in the chronic SIV + ART phase in comparison to acute SIV infection levels are in concordance with the systemic activation of monocytes observed in PWH despite prolonged treatment [34]. Increased plasma concentrations of inflammatory mediators, including IL-6 and MCP-1, can promote monocyte extravasation and subsequent tissue inflammation, as observed with aging [35]. Moreover, elevated levels of sCD14, IL-6, and TNF-α in HIV infection reveal chronic monocyte activation and are likely a major factor for the increased risk of CVD and neurocognitive complications in PWH [36,37]. Accordingly, we found increased frequencies of intermediate (CD14^+^CD16^+^) monocytes in SIV-infected macaques during acute SIV infection before ART as well as chronic infection with ART suppression. The percentage of intermediate monocytes increases with HIV and SIV infection and correlates with the incidence of neuropathology and CVD with HIV and SIV infection [38,39,40]. The nearly significant correlation between circulating intermediate monocyte frequencies and plasma IFABP levels in the relatively small number of macaques in our study validates the interrelation between gut barrier disruption and chronic innate immune activation during prolonged ART. Given that increased microbial translocation as measured by LPS and sCD14 is characteristic of HIV infection even with suppressed viral replication and has been associated with cardiometabolic risk factors [41,42,43], it is likely a major driver of inflammatory monocyte activation. Further, in the SIV-macaque model of HIV-associated CVD and HIV-associated neurocognitive disorder (HAND), plasma IL-18, in addition to galectin-3 and galectin-9, was shown to correlate strongly with monocyte activation and turnover [44]. We believe that if the duration of SIV + ART continued for more than a year in macaques, significant associations with inflammatory monocyte activation would be observed, particularly in gut and cardiovascular tissues.

High cholesterol and elevated TG may contribute to atherogenesis and CVD burden among PWH. Although we did not observe significant increases in blood cholesterol levels during the 1-year SIV + ART period in our study, our findings of a steady increase in TG levels from 5 months onwards is in concordance with the decreased clearance of triglycerides and increase in disease burden observed among PWH with several years of suppressed viral infection [24,45]. Further, our results showing increased glycolysis in PBMCs during chronic SIV infection with long-term ART reflect the elevated glycolysis in HIV-infected CD4 T cells, monocytes, and macrophages to generate metabolites supporting synthesis of lipids, nucleotides, and viral proteins necessary for HIV replication and latency [46,47]. Although our experiments were conducted on unfractionated PBMCs, overall, these findings suggest metabolic reprogramming in immune cells during long-term ART-suppressed viral infection that could contribute to pathogenesis of inflammatory diseases. One important limitation of this study is the relatively small sample size and the lack of data from tissues. We recommend further research focused on inflammasome activation and metabolic function in tissue monocytes and macrophages at precise time points during SIV infection with ART to reveal the mechanistic pathways. Another limitation is the lack of an ART-only group for comparison, since the long-term use of ART may be linked to metabolic abnormalities such as dysregulation of glucose and lipid metabolism depending upon the classes of drugs used [48]. Future studies comparing the inflammasome activation and metabolic function in tissue monocytes and macrophages between SIV + ART and ART-only groups of macaques will provide insights into the precise mechanisms driven by long-term ART-exposure in the presence or absence of virus infection.

In summary, our findings demonstrate a potential link between gut barrier disruption, inflammasome activation, and metabolic remodeling in SIV-infected macaques on ART despite viral suppression. The significant correlations between circulating IL-18, IL-1β, and IFABP suggest gut-derived inflammasome activation as a contributing factor. Furthermore, elevated markers of monocyte activation, triglycerides, and glycolysis indicate a metabolic shift potentially accelerating CVD development. Future research is warranted to explore whether a Western/high-fat diet accelerates HIV-associated cardiovascular pathogenesis in rhesus macaques with chronic SIV infection under long-term suppressive ART. Additionally, it is important to elucidate the mechanisms by which gut dysfunction and inflammasome activation contribute to HIV-associated CVD. This knowledge could lead to targeted interventions to prevent these complications in PWH.

## 4. Materials and Methods

### 4.1. Animals, Viral Inoculation, and ART

Six healthy female Indian-origin rhesus macaques ranging in age from 5 to 10 years old and confirmed to be seronegative for SIV, HIV-2, STLV-1 (Simian T Leukemia Virus type-1), SRV-1 (type D retrovirus), and herpes-B viruses were used in this study. MHC-1 genotyping for exclusion of the common Mamu alleles Mamu-A*01/-A*02 and Mamu-B*08/-B*17 was performed by sequence-specific priming PCR. Animals were infected with 2500 TCID50 SIVmac251 via the intrarectal (IR) route using the pathogenic SIV challenge stocks obtained from the Preclinical Research and Development Branch of Vaccine and Prevention Research Program, Division of AIDS, NIAID. cART consisted of daily subcutaneous injection of 5.1 mg/kg Tenofovir Disoproxil Fumarate (TDF), 30 mg/kg Emtricitabine (FTC) and 2.5 mg/kg Dolutegravir (DTG) in a solution containing 15% (*v/v*) kleptose at pH 4.2, as previously described [49].

### 4.2. Isolation of Peripheral Blood Mononuclear Cells (PBMCs) and Flow Cytometry

Blood samples collected in EDTA vacutainer tubes (Sarstedt Inc., Newton, NC, USA) were processed immediately. PBMCs were separated by density gradient centrifugation (Lymphocyte Separation Medium; MP Biomedicals Inc., Solon, OH, USA) at 1500 rpm for 45 min and used for phenotyping and in vitro functional assays. Multi-color flowcytometric analysis was performed on cells according to standard procedures using anti-human mAbs that cross-react with rhesus macaques. Freshly isolated or frozen PBMCs were stained for flow cytometry as previously described [10]. Briefly, 1–2 million cells were resuspended in 100 μL stain media (PBS with 2% FBS) and incubated with appropriate surface-antibody cocktails for 30 min at 4 °C and washed with Running Buffer. For monocyte immunophenotyping analysis, PBMCs were stained with the antibodies for CD1c (clone BDCA-1), CD3 (clone SP34-2), CD8 (clone SK-1), CD123 (clone 7G3), CD11b (clone ICRF44), CD14 (clone M5E2), CD16 (clone 3G8), CD20 (clone 2H7), and HLA-DR (clone L243). All mAbs utilized in this study were obtained from either BD Biosciences (Franklin Lakes, NJ, USA), Biolegend (San Diego, CA, USA), or Miltenyi Biotec (Teterow, Germany). Stained samples were resuspended in PBS containing 2% paraformaldehyde, and the results were acquired within 24 h after sample processing on BD LSR Fortessa. Unstained samples were run with every set of samples. Analysis of the acquired data was performed using FlowJo software (version 10.9.0; TreeStar (Woodburn, OR, USA)). Monocytes were identified as HLA-DR^+^CD3^−^CD20^−^CD8^−^ cell populations that were further divided as (a) CD14^+^CD16^−^ classical monocytes, (b) CD14^+^CD16^+^ intermediate monocytes, and (c) CD14^lo/neg^CD16^++^ nonclassical monocytes, as previously described [50].

### 4.3. Plasma Markers of Inflammation, Microbial Translocation, and Intestinal Damage

Frozen plasma samples were thawed and cleared using Ultrafree Centrifugal Filters (Millipore, Billerica, MA, USA). The filtered plasma samples were used for simultaneous quantification of cytokines, chemokines, and growth factors using the multiplexed-bead assay Non-Human Primate Cytokine & Chemokine & Growth Factor 37-plex ProcartaPlex (Invitrogen (Waltham, MA, USA), Life Technologies (Carlsbad, CA, USA)), according to the manufacturers’ instructions. Data were acquired with a Bio-Plex 200 analyzer (Bio-Rad, Hercules, CA, USA) and analyzed using Bio-Plex Manager software v6.1 (BioRad, Hercules, CA, USA). For the analysis of markers of leaky gut and microbial translocation, plasma IFABP and LBP were quantified using the commercially available Monkey IFABP/FABP2 and LBP ELISA kits (MyBioSource, San Diego, CA, USA). A commercially available ELISA kit for Human sCD14 (R&D Systems, Minneapolis, MN, USA) was used according to the manufacturer’s protocols with 1:200 diluted plasma samples. All tests were performed according to the manufacturer’s guidelines. The assays were performed in duplicate, and data were analyzed using Gen 5 software (BioTek, Winooski, VT, USA).

### 4.4. Measurement of Caspase-1 in Serum

Caspase-1 levels in macaque serum samples were quantitatively determined using the Quantikine™ ELISA Human caspase-1/ICE Immunoassay (Catalog# DCA100, R&D Systems, Minneapolis, MN, USA). The assay was conducted according to the manufacturer’s instructions. Serum samples were centrifuged at 3000× *g* for 15 min to remove particulates. Samples were stored at −80 °C until analysis to prevent degradation. The plate provided in the kit was pre-coated with a monoclonal antibody specific to the p20 subunit of human caspase-1. Samples and standards were pipetted into the wells, and any caspase-1 present was bound by the immobilized antibody. After washing away unbound substances, a polyclonal antibody specific for human caspase-1 was added, followed by a horseradish peroxidase (HRP)-conjugated anti-rabbit IgG antibody. After the washing step, a substrate solution was added, and the color developed in proportion to the amount of caspase-1 bound. The reaction was terminated by the addition of a sulfuric acid stop solution, and the optical density was measured at 450 nm. Data were analyzed using the standard samples provided with the kit.

### 4.5. Measurement of Glycolytic Rate and Extracellular Acidification Rates (ECARs)

ECAR measurements were performed using a glycolysis stress kit and SeaHorse XF24 analyzer (Agilent, Santa Clara, CA, USA). The assay was conducted according to the manufacturer’s instructions. Briefly, PBMCs from control SIV-naïve and SIV-infected + ART-treated macaques were plated into XF24 polystyrene cell culture plates (SeaHorse Bioscience, North Billerica, MA, USA) at 20,000/well. Both groups were either untreated or treated with Ox-LDL (10 μg/mL) for 24 h and glycolysis measured using the glycolysis stress kit and SeaHorse XF24 analyzer. The data were plotted using Prism 10 software and the mean of 5 replicates from 2 animals in each group was used for analysis.

### 4.6. Statistical Analyses

All statistical analysis was performed using GraphPad Prism Software (Version 10.0.1). Data were analyzed by one-way analysis of variance (ANOVA) with multiple comparisons, one-way ANOVA with a test for linear trend, or two-way ANOVA with repeated measures. Tukey’s and Dunnett’s post hoc tests were used for multiple comparisons. All correlations were computed using a non-parametric Spearman rank correlation test. *p* values of 0.05 or lower were considered significant, * *p* < 0.05, ** *p* < 0.01, *** *p* < 0.001. Data were analyzed by using Student’s *t*-test for comparison of glycolytic capacity between SIV-naïve and SIV-infected macaque groups.

## Figures and Tables

**Figure 1 ijms-25-08702-f001:**
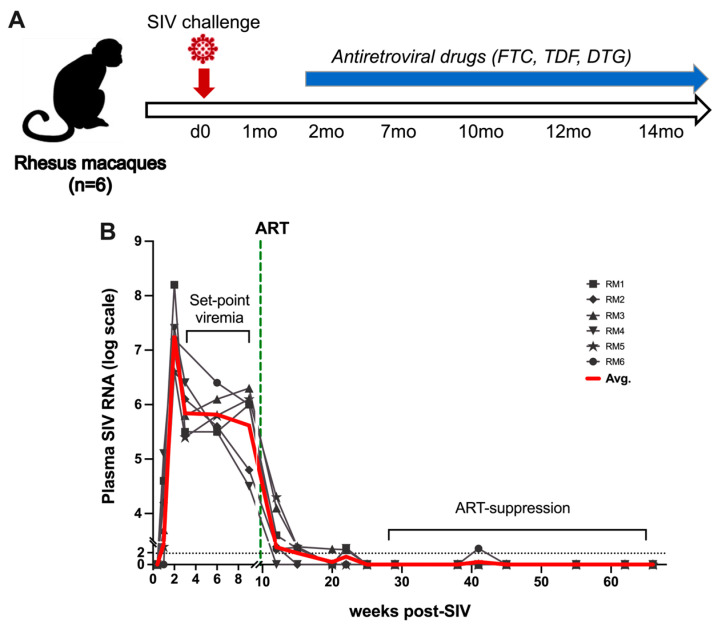
Study design and viral loads. (**A**) Six rhesus macaques (RM) were inoculated with SIV and treated with combination ART consisting of daily subcutaneous injection of Tenofovir Disoproxil Fumarate (TDF), Emtricitabine (FTC), and Dolutegravir (DTG). (**B**) SIV RNA quantification in plasma over 14 months of infection in the study. The dashed line represents the beginning of ART. Each symbol represents an animal and the red line is the average viral load for all animals.

**Figure 2 ijms-25-08702-f002:**
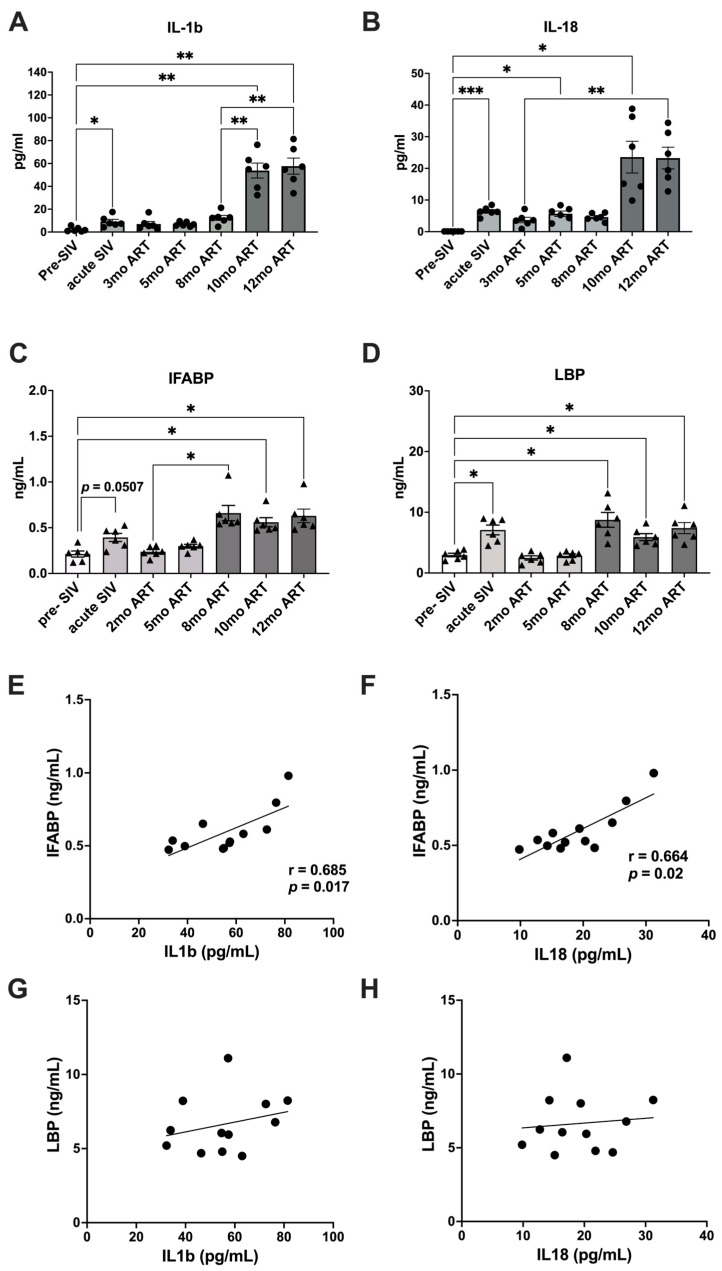
Plasma levels of inflammasome/caspase-1 pathway activation and IEBD biomarkers. Longitudinal plasma levels of inflammasome activated cytokines IL-1β (**A**) and IL-18 (**B**) measured by Luminex assay in SIV-infected rhesus macaques. Plasma levels of IFABP (**C**), a marker of enterocyte loss and generalized damage to the intestinal epithelium and LBP (**D**), a marker of host response to LPS via microbial translocation measured at serial time-points pre-SIV baseline and post-SIV + ART. Two technical replicates were used for each time-point. One-way ANOVA with Tukey’s multiple comparisons test was used to determine significant differences between baseline and different time points post-SIV infection and ART. Asterisks indicate significant differences between time points (* *p* < 0.05; ** *p* < 0.01; *** *p* < 0.001). Associations of IFABP with IL-1β (**E**) and IL-18 (**F**) showing significant positive correlations during 10–12 months of ART. Non-significant correlations between LBP and IL-1β (**G**) and IL-18 (**H**) in the bottom panel. Statistical correlations were investigated by the Spearman correlation coefficient and 95% confidence limits. Statistical significance (*p* values) and Spearman’s coefficient of rank correlation (r) are shown for significant correlations.

**Figure 3 ijms-25-08702-f003:**
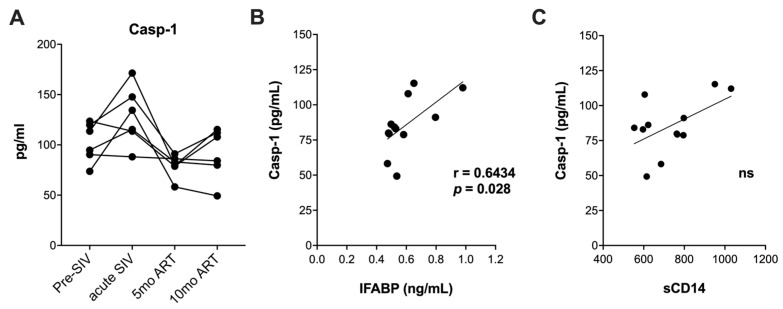
Circulating levels of caspase-1 and associations with IEBD biomarkers. (**A**) Longitudinal serum levels of caspase-1 (Casp-1) measured by ELISA. Associations of caspase-1 with IFABP (**B**) and sCD14 (**C**) showing significant positive correlation for IFABP during 10–12 months of ART. Statistical significance (*p* value) and Spearman’s coefficient of rank correlation (r) are shown for significant correlation.

**Figure 4 ijms-25-08702-f004:**
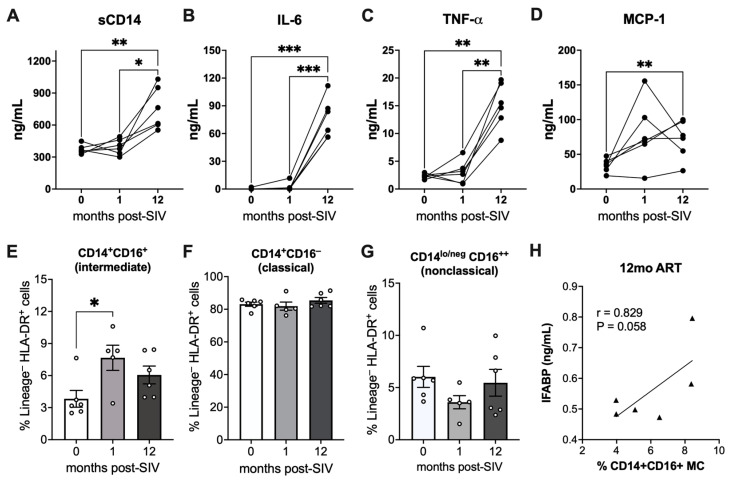
Circulating levels of inflammatory cytokines and frequencies of monocyte subpopulations in PBMC. Plasma levels of sCD14 (**A**), IL-6 (**B**), TNF-*α* (**C**), and MCP-1 (**D**) at pre-SIV, 1 month post-SIV and 12 months post-SIV infection. Asterisks indicate significant differences between time-points (* *p* < 0.05; ** *p* < 0.01; *** *p* < 0.001). Frequencies of circulating monocyte subsets comprising CD14^+^CD16^+^ intermediate monocytes (**E**), CD14^+^CD16^−^ classical monocytes (**F**), and CD14^lo/neg^CD16^++^ nonclassical monocytes (**G**) in PBMC at the 0-, 1-, and 12-month post-SIV time-points. (**H**) Association between plasma IFABP levels and frequencies of intermediate monocyte at 12 months of ART. Statistical significance (*p* value) and Spearman’s coefficient of rank correlation (r) are shown.

**Figure 5 ijms-25-08702-f005:**
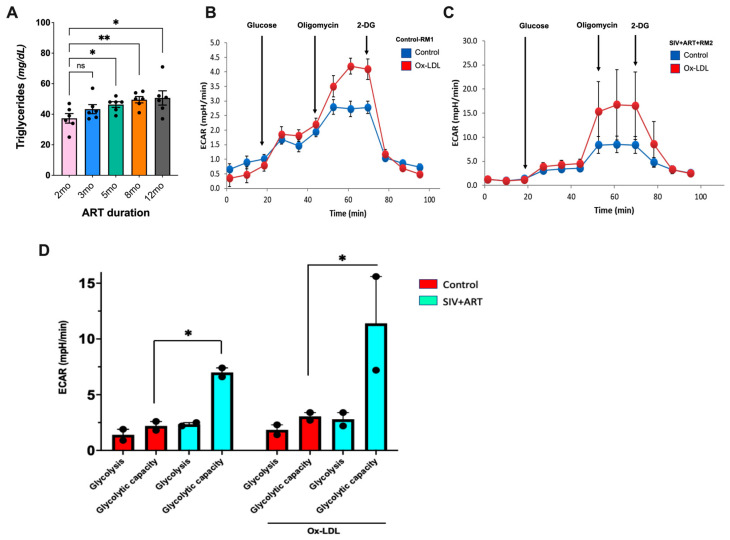
Circulating levels of triglycerides and PBMC glycolytic capacity during chronic SIV-infection under long-term virus-suppressive ART. (**A**) Triglyceride (TG) levels evaluated by blood chemistry on serial blood draws from 2 to 12 months of ART in SIV-infected macaques (* *p* < 0.05, ** *p* < 0.01; ns, not significant). (**B**,**C**) Representative extracellular acidification rate (ECAR) measurements in one SIV-naïve control rhesus macaque (RM1) and one SIV + RM at 12 months of ART. (**D**) Increased glycolytic capacity in Ox-LDL-treated and -untreated PBMCs from SIV-infected ART-treated RMs (blue bars) in comparison to Ox-LDL-treated and -untreated PBMCs from SIV-naïve RMs (red bars). The extracellular acidification rate (ECAR) was measured in 0.2 × 10^5^ PBMCs from both groups that were left untreated or treated with Ox-LDL (10 μg/mL) for 24 h. Each dot represents the mean of 5 replicates from 2 animals. Asterisk (*) represents *p* < 0.05 significant differences determined using Student’s *t*-test.

## Data Availability

All the data obtained during this study are included in the manuscript. Additional information can be provided by the authors upon reasonable request.

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
