# Peer review of "The Association between IL-1β and IL-18 Levels, Gut Barrier Disruption, and Monocyte Activation during Chronic Simian Immunodeficiency Virus Infection and Long-Term Suppressive Antiretroviral Therapy"

_ijms, 2024, doi:10.3390/ijms25168702_

Round 1

Reviewer 1 Report

Comments and Suggestions for Authors

Findings suggest that gut barrier disruption correlates with inflammasome activation during chronic SIV infection despite long-term virus-suppressive ART. This is associated with changes in intermediate monocyte subpopulations and increased proinflammatory cytokines as well as elevated triglycerides in blood. Investigators demonstrated a significant correlation between the increased IEBD biomarker IFABP and circulating inflammasome byproducts, IL-1β and IL-18 during chronic SIV infection with long-term suppressive ART. Furthermore, authors showed increased plasma markers of monocyte activation, along with elevated blood triglyceride levels and enhanced glycolytic activity in PBMCs of macaques with long-term SIV+ART. Importantly, higher increases in IL-1β and IL-18 levels were observed in the SIV-infected macaques at 10-12 months of ART (6-fold higher IL-1β levels and 3.6-fold higher IL-18 levels )than the levels observed during acute SIV infection) indicative of more robust and persistent inflammasome activation during long-term SIV+ART. Study is well done and most parts supports already known phenomenon I chronic immune activation despite ART, therefore incrementally increasing knowledge in this area. Several issues need to be addressed.

1.     There is no ART group alone without SIV infection. ART is known to mediate metabolic effects, yet this issue is completely ignored in current study.

2.     While marker of increased permeability correlate will with pro-inflammatory cytokines, microbial traslocation seems to play a minor role (at least with employed markers). How authors explain this fact and what is driving force behind immune activation?

3.     If indeed monocytes play significant role in CVD should functional assays should be attempted (migration, adhesion, etc.)?

Author Response

Comment 1: There is no ART group alone without SIV infection. ART is known to mediate metabolic effects, yet this issue is completely ignored in current study.

Response 1: We appreciate the reviewer’s comment and agree that long-term exposure to ART itself can mediate metabolic effects, and that an ART only group would be the most appropriate control group for this study. Indeed, in future studies, we plan to leverage samples from other longitudinal studies with a similar design to further compare our results. However, the nature of this prospective, longitudinal, single-arm study of chronic SIV infection with long-term ART (> 1 year) in rhesus macaques precluded us from including a similar comparison group of ART-only treated animals in this current study. This study design is common in nonhuman primates owing to logistical and ethical issues and to minimize use of this acutely scarce resource. Further, we believe that the strengths of the rigorous multiple baseline data matching improving the statistical analysis in terms of power calculations, and use of matching information in the statistical testing of the longitudinal/time-course effects (post-infection/treatment period) outweigh the limitation of lack of a comparison with ART-only group. To address the reviewer’s important point, in the discussion section (lines 361-367), we have now added the lack of an ART-only group as one of the limitations of this study design.

Comment 2:     While marker of increased permeability correlate will with pro-inflammatory cytokines, microbial translocation seems to play a minor role (at least with employed markers). How authors explain this fact and what is driving force behind immune activation?

Response 2: The markers of microbial translocation employed in this study included LPS-binding protein (LBP) and sCD14. While our results showed an increase over baseline levels for both LBP and sCD14 during long-term SIV+ART, we found no significant correlations between these markers and IL-1b, IL-18, or Casp-1, unlike the correlations with the marker of intestinal epithelial damage (IFABP). This led us to hypothesize that IEBD is a major driving mechanism behind immune activation and the activation of inflammasome pathway. While microbial translocation is a consequence of IEBD and may further exacerbate inflammation, LBP and sCD14 are indirect markers of microbial translocation, the production of which is induced by the presence of LPS and activation of monocytes. This underscores the fact that there may be other contributing factors to monocyte activation during chronic SIV infection and long-term ART that may have impacted the direct correlation analyses between LBP and sCD14 and inflammasome activation markers in this study.

Comment 3:     If indeed monocytes play significant role in CVD should functional assays should be attempted (migration, adhesion, etc.)?

Response 3: We completely agree with this point that functional assays to investigate monocyte functions including migration, adhesion assays, etc. would be important in revealing the cellular mechanisms involved. Indeed, we are embarking on follow up studies to determine the role of monocyte activation in tissue-specific inflammasome activation in the context of HIV/SIV-associated CVD. However, it is beyond the scope of this current manuscript that focuses on the associations of IEBD in driving inflammation and monocyte activation.

Reviewer 2 Report

Comments and Suggestions for Authors

The manuscript looks at the effect of SIV infection on inflammatory cytokines in rhesus macaques. The results indicate higher plasma levels of IL-1β and IL-18, increase  intestinal fatty acid-binding protein and LPS-binding protein during the chronic phase of treated SIV infection

the results are appropriate, but the major concern is the overinterpretation of results.

Though ILb are indicators of inflammasome activation, the study does not provide evidence of inflammasome (NLRP3 for eg or the like). so please rephrase.

the whole premise of the manuscript is the potential increase in CVD, but there is no indication of CVD or markers of CVD, please rephrase

Triglyceride level is in the normal clinical range , correct?

the seahorse results - typically atleast 2X105 cells are needed for PBMCs for good results. moreover, interpretation from the data from 2 animals is not sufficient to make conclusions. Either more numbers or removing this data is recommended. It does not anyway add a lot to the objective of the study. 

It is recommended that the authors rephrase the intro and discussion to show that there are changes in inflammatory and IEBD markers. And rephrase that they are indication of CVD.

Comments on the Quality of English Language

No comments, the text is good

Author Response

Comment 1: Though ILb are indicators of inflammasome activation, the study does not provide evidence of inflammasome (NLRP3 for eg or the like). so please rephrase. The whole premise of the manuscript is the potential increase in CVD, but there is no indication of CVD or markers of CVD, please rephrase.

Response 1: We appreciate the reviewer’s comments and have made changes accordingly in the abstract and elsewhere to address this concern. We agree that in our study there was no assessment of direct markers of CVD or evaluation of the NLRP3 inflammasome pathway upstream of IL-1b and IL-18 production. We have rephrased the abstract to replace CVD pathogenesis with metabolic dysfunction.

Comment 2: Triglyceride level is in the normal clinical range, correct?

Response 2: Regarding the triglyceride levels, it is challenging to estimate the normal clinical range in macaques as they display a wide range, and the range for metabolic disease has not been determined. One study reported adult males: 7.40–134; adult females: 19.0–78.0 md/dL; Koo et al., PMID: 32257895. While another study reported 41 ± 22 md/dL in 9-17 years old adult females; Wurz et al., https://doi.org/10.1111/j.1742-7843.2008.00235.x, which is more representative of our study cohort comprising all female rhesus macaques. Although the triglyceride levels did not fall outside this range in the study animals even during chronic infection with long-term ART, we aim to highlight the steady increase over time in all animals without any diet intervention. We speculate that over a longer duration or with added factors like high fat/high fructose diet (western diet), the increase may contribute to hypertriglyceridemia and metabolic disease.

Comment 3: the seahorse results - typically at least 2X105 cells are needed for PBMCs for good results. moreover, interpretation from the data from 2 animals is not sufficient to make conclusions. Either more numbers or removing this data is recommended. It does not anyway add a lot to the objective of the study. 

Response 3: According to the manufacturer (Agilent) website, “robust response with as few as 10,000 cells per well in the custom 24-well plate” can be detected. "https://www.agilent.com/en/product/cell-analysis/real-time-cell-metabolic-analysis/xf-analyzers/seahorse-xfe24-analyzer-740878. The readout of this assay is proportional to cell numbers used. We had no issues obtaining results from the cell numbers used in our assay.

Even with n=2/group, we observed a significant difference in glycolytic potential suggesting that SIV infection with long-term ART induces changes in cellular metabolic activity in comparison to SIV-naïve animals. Thus, we would like to retain the data in the figure. This adds to the objective of the study to understand how intestinal epithelial barrier disruption during chronic SIV+ART, and inflammasome-related cytokines IL-1b and IL-18 are connected and may be associated with metabolic dysfunction in immune cells.

Comment 4: It is recommended that the authors rephrase the intro and discussion to show that there are changes in inflammatory and IEBD markers. And rephrase that they are indication of CVD.

Response 4: We have removed major portions about CVD in the introduction section and rephrased the discussion to show that the changes in inflammatory and IEBD markers may be contributing factors for CVD pathogenesis in the long-term, as in the clinical scenario of multiple years of ART-suppressed HIV infection.

Round 2

Reviewer 2 Report

Comments and Suggestions for Authors

Thank you for the responses. Can you please include representative traces of ECAR in the figure. If you looked at published literature, many manuscripts use (PMC8897018, PMC9658277) 2x105 cells. not sure how stats are even done with 2 animals data. Recommended more data be shown for the seahorse analysis. including OCR and ECAR curves.  

Author Response

Comments: Can you please include representative traces of ECAR in the figure. If you looked at published literature, many manuscripts use (PMC8897018, PMC9658277) 2x105 cells. not sure how stats are even done with 2 animals data. Recommended more data be shown for the seahorse analysis. including OCR and ECAR curves.

Response 1: We thank the reviewer for this suggestion. Based on their recommendation, we have now included the representative ECAR data in the updated Figure 5. We have also included the ECAR figures for each animal in the attached figure.

We agree that most manuscripts in the published literature have used 2x105 cells, but we have routinely obtained reliable data from 2x104 cells, and it is well above the lower range of 10,000 cells per well based on manufacturer recommendations.

It is acceptable to perform a t-test with just two samples in each group (https://www.graphpad.com/support/faqid/591/). While having more data increases the statistical power and reduces Type II errors, a sample size of n=2 in our experiment was still sufficient for obtaining valid results. In our study, the rigor is further strengthened by using 5 replicates for each sample (each dot in the figure represents the mean of 5 replicates from each animal). We plan to conduct future experiments with larger sample sizes and more focused cell-specific analyses. However, for the current manuscript and considering time constraints, we intend to keep the analysis at n=2 per group.

Round 3

Reviewer 2 Report

Comments and Suggestions for Authors

No further comments. 

Graphpad allows for doing statistical analysis for 2 samples, does not mean that it is correct.